# UNSUPERVISED ASR
# VIA CROSS-LINGUAL PSEUDO-LABELING

## ABSTRACT

Recent work has shown that it is possible to train an *unsupervised* automatic speech recognition (ASR) system using only unpaired audio and text. Existing unsupervised ASR methods assume that no labeled data can be used for training. We argue that even if one does not have any labeled audio for a given language, there is *always* labeled data available for other languages. We show that it is possible to use character-level acoustic models (AMs) from other languages to bootstrap an *unsupervised* AM in a new language. Here, "unsupervised" means no labeled audio is available for the *target* language. Our approach is based on two key ingredients: (i) generating pseudo-labels (PLs) of the *target* language using some *other* language AM and (ii) constraining these PLs with a *target language model*. Our approach is effective on Common Voice: e.g. transfer of English AM to Swahili achieves 18% WER. It also outperforms character-based wav2vec-U 2.0 by 15% absolute WER on LJSpeech with 800h of labeled German data instead of 60k hours of unlabeled English data.

Table 1: Key idea of the paper: starting from an acoustic model for some other *source* language (e.g. Spanish, $es \rightarrow$) and generating pseudo-labels using a language model for the desired *target* language (e.g. English), we can train an unsupervised speech recognition system for the *target* language using iterative pseudo-labeling. In the example below, we show the ground truth transcription for an English audio and the evolution of its pseudo-labels for our method: by the end of training we are able to reconstruct most of the words in the ground truth transcription.

| iteration | this requires more insulators and wire but doubles the power without doubling the poles |
|---|---|
| $es \rightarrow$ | dister qiris more ance latters a mater ot tobus of pa o tou tholin na pos |
| 0-4k | destroy arise more ance later and water tables of pa to the line pos |
| 4k-8k | desert cars more in later a water toes apart doing pas |
| 8k-12k | this requires more in later and later to apart doing pas |
| 12k-16k | this requires more in later and later does the part doubling pos |
| 16k-20k | this requires more in later and water doubles the part doubling pos |
| 20k-24k | this requires more in later and water double the power with doubling pos |
| 24k-28k | this requires more insulator and water double the power with doubling past |
| 28k-32k | this requires more insulators and water double the power with doubling past |

## 1 INTRODUCTION

How many hours of labeled audio do we need to train a good automatic speech recognition (ASR) system? Over the past several years the answer has been steadily decreasing, and might even be "zero". Recent research has shown that large unlabeled audio datasets can be harnessed to train state-of-the-art acoustic models (AMs).

The two dominant methods for leveraging unlabeled audio are unsupervised pre-training via self-supervision (SSL) (Baevski et al., 2020; Hsu et al., 2021; Chung & et al., 2021; Baevski et al., 2022) and semi-supervised self-training (Kahn et al., 2020; Xu et al., 2020; Likhomanenko et al., 2021; Manohar & et al., 2021; Higuchi et al., 2021b; 2022), or pseudo-labeling (PL). In pre-training, a model is trained to process the raw unlabeled data to extract features that solve some pretext task, followed by supervised fine-tuning on some downstream ASR task. In pseudo-labeling, a model is used to generate pseudo-labels (PLs) for the unlabeled data, and standard supervised training methods

| Source: English (EN)  Target: Swahili (SW) | SW ground truth: | kamwe vilio havizuii jambo |
|---|---|---|
| | EN AM + EN LM: | come maybe little havizutamba |
| | EN AM: | kam me vileo havizui jamba |
| | EN AM + SW LM: | kamwe vilio havijui jambo |

Figure 1: Motivation: reasonable zero-shot ASR for Swahili is possible by decoding with an English acoustic model (AM) constrained by a Swahili language model (LM), suggesting that training on the resulting pseudo-labels could improve the acoustic model.

can then ingest this pseudo-labeled data, in lieu of regular labeled data. Pre-training and self-training have been found to be complementary (Xu et al., 2021; Zhang & et al., 2022; Berrebbi et al., 2022b), and the combination yields results competitive with purely supervised systems, using only a small fraction of labeled data. Both pre-training and pseudo-labeling assume the existence of at least some amount of labeled audio for training a supervised model. However, for most of the world's roughly 7000 languages (Lewis et al., 2009), there is no labeled audio available.

In this work, we explore an unsupervised setting where only unpaired audio and text data is available for some *target* language. We also assume the availability of labeled data for some *source* language. Our motivation (see Figure 1) lies in the practical observation that a lot of paired data is already freely available for some high resource languages, such as English (e.g. there are ∼3k hours of transcribed English in Common Voice (Ardila et al., 2020)). We show that an acoustic model trained on the *source* language can perform cross-lingual transfer to the *target* language, without any labeled audio from the *target* language. Unlike much previous work, our method does not depend on adversarial training or phonetic lexicon information, and instead uses simple end-to-end character-level acoustic models and standard self-training recipes commonly used in ASR. Cross-lingual transfer is performed via unsupervised cross-lingual PL. Assuming some *target* language audio is available, the core idea of our approach is to generate PLs with the *source* language acoustic model fed by the *target* audio. These PLs are further constrained by a language model (LM) on the *target* language. We show this approach generates good enough PLs to bootstrap an unsupervised *target* language acoustic model, assuming languages are from the same family group. In fact, even for languages from different language family groups, we show unsupervised ASR via cross-lingual PL is promising: e.g. with English as a *source* we achieve 23.7% (18%) word error rate on Swahili as *target* language, using greedy (LM beam-search) decoding.

## 2 BACKGROUND

Unsupervised ASR (Aldarmaki et al., 2022) attempts to train an acoustic model (AM) using only unpaired audio and unpaired text. Substantial progress in unsupervised ASR was recently made by wav2vec-U (Baevski et al., 2021; Liu et al., 2022), building on top of strong self-supervised (SSL) representations from wav2vec (Baevski et al., 2020): these models apply adversarial learning to automatically learn a mapping between audio representations and token units (either characters or phonemes). Reported experiments show a significant gap between phoneme-based (text is phonemized) and character-based approaches, which led the authors to conclude that "end-to-end unsupervised ASR could be possible but further research and development are necessary". ASR2K (Li et al., 2022) investigates an ASR system which does not require any acoustic data for the *target* language. Instead, a multilingual phone-based supervised ASR model is trained on several *source* languages, and combined with a G2P (grapheme-to-phoneme) model. Then, given a *target* language, the corresponding acoustic model is inferred by matching the supervised acoustic model output statistics to available $n$-gram statistics (computed on a *target* text-only corpus). For *target* languages that the G2P model has not seen at training time, an ensemble is performed over G2P predictions from nearest languages. More generally, previous works on multilingual ASR (not necessarily unsupervised) showed that languages may share similar representations, and that data from another language may thus improve overall performance on a *target* language (Ghoshal et al., 2013; Pratap et al., 2020; Lugosch et al., 2022; Chen et al., 2022; Thomas et al., 2020). As for ASR2K, our approach relies on labeled data from some source languages as initial audio representations. The most closest work to ours (Klejch et al., 2022) exploits self-training idea though it requires a universal multilingual phonemizer: if you have a phoneme-based lexicon for the target language, the

unsupervised ASR problem is virtually solved by using e.g. ASR2K. However, in contrast to both ASR2K and wav2vec-U, we show that end-to-end (character-based) unsupervised ASR is viable, as long as source and target languages share enough "similarities". In that respect, in the hope to shed some light on the importance of language similarities, most experiments of this paper consider only pairs of (one source, one target) languages. Experiments in Section 5.5 show that a multilingual source AM improves ASR on target languages. In Section 5.6, we further extend the approach to a setting where the source and target languages have different alphabets. The following subsections now formally introduce the way our acoustic models and language models are trained, as well as the classical (monolingual) pseudo-labeling approach.

## 2.1 ACOUSTIC (AM) AND LANGUAGE (LM) MODELS

Let $\mathbf{x}$ denote the audio and $\mathbf{y}$ denote its transcript. End-to-end transformer-based AMs trained with Connectionist Temporal Classification (CTC) (Graves et al., 2006) loss showed (Higuchi et al., 2021a; Burchi & Vielzeuf, 2021; Kim et al., 2022) a good performance when decoded greedily by picking the most probable token for each frame and removing repetitions and blanks. However, greedy decoding does not always find the best $\mathbf{y}$, as CTC marginalizes over alignments leading to the ground truth transcription (and does not simply consider the best alignment). Finding the best $\mathbf{y}$ according to CTC loss would require an intractable exhaustive search. Instead, approximation is performed with a beam search procedure, which maintains and updates a limited number of hypotheses. During this procedure, the search can be constrained to words from a lexicon. In addition, it is often beneficial to involve an LM trained on the external text to out-weight more plausible transcript hypotheses. Simple $n$-gram LMs can be integrated efficiently into beam-search decoding and show improved results. Formally, an LM beam search aims at maximizing:

$$\max_{\mathbf{y}} \ \log p^{\mathrm{AM}}(\mathbf{y}|\mathbf{x};\theta) + \alpha \log p^{\mathrm{LM}}(\mathbf{y}) + \beta|\mathbf{y}|, \tag{1}$$

where $\theta$ are the trainable parameters of the acoustic model. The hyper-parameters $\alpha, \beta \in \mathbb{R}$ control the reliance on the LM and the number of words $|\mathbf{y}|$ in $\mathbf{y}$, respectively.

## 2.2 MONOLINGUAL PSEUDO-LABELING (PL)

Let $D_L$ denote a labeled set of audio-text pairs $(\mathbf{x}, \mathbf{y})$ and $D_U$ an unlabeled set of audio $\mathbf{x}$ only. Supervised training minimizes the conditional negative log-likelihood $\min_{\theta} \ \mathbb{E}_{\mathbf{x},\mathbf{y}\sim D_L} \ -\log p^{\mathrm{AM}}(\mathbf{y}|\mathbf{x};\theta)$, on the labeled data $D_L$, while pseudo-labeling additionally minimizes its counterpart on the unlabeled data $D_U$, $\min_{\theta} \ \mathbb{E}_{\mathbf{x}\sim D_U} \ -\log p^{\mathrm{AM}}(\hat{\mathbf{y}}|\mathbf{x};\theta)$, where $\hat{\mathbf{y}}(\mathbf{x}) = \mathrm{PL}(\mathbf{x};\theta')$ denotes a pseudo-label (PL) transcription generated by some method. Different ways of inferring pseudo-labels $\mathrm{PL}(\mathbf{x};\theta')$ have been proposed (Kahn et al., 2020; Park et al., 2020; Xu et al., 2020; Likhomanenko et al., 2021; Manohar & et al., 2021; Higuchi et al., 2022; Berrebbi et al., 2022a), including both greedy and beam-search decoding, with or without an external LM, and with variants on the "teacher" AM model $\theta'$. IPL (Xu et al., 2020) and slimIPL (Likhomanenko et al., 2021) are continuous PL approaches, where a single AM (with parameters $\theta$) is continuously trained. At each iteration one uses either labeled data (to ground the model), or unlabeled data with pseudo-labels $PL(\mathbf{x};\theta')$ obtained via the maximization shown in Equation (1), with an older version $\theta'$ of the AM. Picking a previous version of the AM $\theta'$ has been shown to help stabilize the procedure. IPL relies on LM beam-search decoding ($\alpha > 0$) for PLs, which gives stable training but can lead to over-fitting to the language model. Using greedy decoding is more efficient (as it does not involve a beam-search), and has been shown to lead to better word error rate performance, but can be subject to training instabilities. Derived from IPL, slimIPL is an LM-free approach, which resolves training issues by constraining further the AM teacher model $\theta'$: either model averaging is performed, or PLs are obtained with different past versions of the AM. In the latter case, a cache of PLs is maintained for efficiency, allowing the same PL to be reused several times in the training.

## 3 CROSS-LINGUAL PSEUDO-LABELING

In this work, we consider a different setting than classical PL approaches (see Figure 2), where the *target* language has only *unlabeled* audio data $D_U^{\mathrm{tgt}}$ available. To circumvent the lack of labeled data,

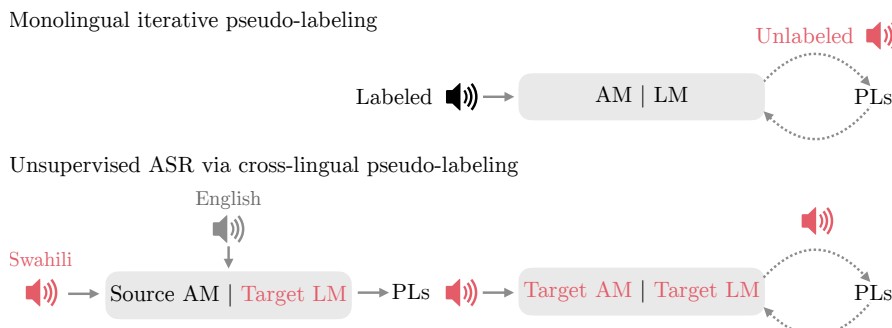

Figure 2: Comparison of standard monolingual pseudo-labeling and unsupervised ASR via cross-lingual pseudo-labeling, where labeled data are available for a *source* language and no labeled audio is available for the *target* language.

we show that it is enough to assign *target* acoustic data with PLs from a different *source* language. The *source* language AM is assumed to be trained with audio labeled data $D_L^{\text{src}}$. As motivated in Figure 1, even though the *source* language AM has never seen *target* audio at training, we show that constraining PL generation with a beam-search decoding (with an LM trained on a *target* text corpus $T^{\text{tgt}}$) is enough to bootstrap a *target* AM.

More formally, we train a *source* language AM on $D_L^{\text{src}}$ and a *target* language LM. We then perform what we call "Cross-Lingual Pseudo-Labeling"[1], to train a *target* language AM on unlabeled audio $D_U^{\text{tgt}}$. It is composed of two phases, the second one being optional:

**Phase 1** A *target* AM is bootstrapped via cross-lingual IPL, as shown in Figure 2:

- The *target* AM is initialized with the *source* AM.
- We perform IPL, with PLs generated via beam-search decoding constrained with the *target* LM.

**Phase 2** A resulting *target* AM is further trained via slimIPL:

- The *target* AM is reinitialized with the *source* AM. It is then fine-tuned with PLs obtained from Phase 1 with a larger beam.
- A regular continuous slimIPL procedure (LM-free) is performed over the *target* AM.

Phase 1 is essential to bootstrap the *target* AM, as Cross-Lingual Pseudo-Labeling would fail (as suggested by decoding results shown in Table 1), if no *target* LM was there to constrain PL generation. While most of the paper results are obtained with Phase 1 only, we show in Section 5.3 that Phase 2 can provide a boost in WER performance. This corroborates findings from the original slimIPL algorithm (Likhomanenko et al., 2021), in monolingual settings.

## 4 EXPERIMENTAL SETUP

We perform a number of experiments using different pairs of languages. Also several language groups are considered, as shown in Table 2. We use the multilingual Common Voice (Ardila et al., 2020) dataset (version v12.0). For *source* languages, we picked available paired audio and text, while for *target* languages only audio data was considered. To train *target* LMs, we picked text data available in Common Crawl data[2] (Wenzek et al., 2020; Conneau et al., 2020). We performed experiments with English, German, Spanish, and French as both *source* and *target* languages. For African languages, we use Kinyarwanda only as a *source* language (as text data are not available in the Common Crawl dataset), and Swahili and Hausa only as *target* languages (as they are low resource). Finally, to compare with wav2vec-U 2.0 (Baevski et al., 2021) we use LJSpeech audio (Ito & Johnson, 2017) as $D_U^{\text{tgt}}$ and LibriSpeech (Panayotov et al., 2015) LM corpus data as $T^{\text{tgt}}$.

---

[1]"Cross-Lingual Pseudo-Labeling" should not be confused with "cross-lingual self-training" from (Zhang et al., 2021), which relates to multilingual representation learning.

[2]https://data.statmt.org/cc-100.

Table 2: Characteristics of the languages (from Common Voice v12.0) referred in our empirical study.

| Language | Notation | Language Group | Alphabet |
|---|---|---|---|
| English | en | Indo-European, West Germanic | Latin |
| German | de | Indo-European, West Germanic[4] | {Latin, öäüß} |
| Spanish | es | Indo-European, Romance | {Latin, áéíóúñüý} |
| French | fr | Indo-European, Romance | {Latin, àâæçéèêëîïôœùûüÿ} |
| Kinyarwanda | rw | Niger-Congo | Latin |
| Swahili | sw | Niger-Congo | Latin |
| Hausa | ha | Afro-Asiatic | {Latin, ɓɗƙyˑ } |

## 4.1 TRAINING DETAILS

**Token Set & Text Normalization** The token set used in all our experiments is composed of the union of all characters available in all source languages (en, de, es, fr, rw) alphabets (see Table 2), augmented with a word boundary token, apostrophe and hyphen, resulting in a total of 54 characters. Common Voice transcriptions and Common Crawl text data were normalized by (i) lower casing; (ii) removing punctuation; (iii) converting characters into the Latin token set via `unidecode`[3] package; characters failing the conversion were discarded from the text.

**Acoustic Model (AM)** We use a 36-blocks Transformer model prefixed with a single convolutional layer (kernel 7, stride 3), as in (Likhomanenko et al., 2021). Inputs are 80-dimensional log Mel filterbanks extracted with 25ms window and 10ms stride. Positions are encoded with an absolute sinusoidal positional embedding (Vaswani et al., 2017) as Common Voice audio durations are short (in average 5s). Dropout is set to 0.1 (0.3 for Swahili). The resulting model has 255M trainable parameters.

**AM Training** All acoustic models are trained with the CTC loss, and the Adagrad optimizer. After a warmup period of 64k iterations, training is performed with a learning rate of 0.03 for up to 300k iterations, on 8 GPUs (A100 40GB), and batch sizes of $\sim$ 290s of audio per GPU. As for data augmentation, we use SpecAugment (Park et al., 2019), following the parameters chosen in (Likhomanenko et al., 2021): 2 frequency masks with 30 mask size, 10 time masks with 50 mask size, and probability 0.1. The best models are selected according to the word error rate (WER) performance on validation sets, and final performance is reported on test sets. For all experiments we report both word (WER) and character (CER) error rates.

## 4.2 SUPERVISED MONOLINGUAL ACOUSTIC MODELS BASELINES

We consider English, German, Spanish, French, and Kinyarwanda as *source* languages. Only Common Voice data was used to train AMs. Performance of all used further monolingual *source* AMs is given in Table 3. In our work, Hausa and Swahili were used only as *target* languages. We were not able to successfully train a transformer AM on Hausa (2.3h of training data).

## 4.3 MONOLINGUAL LANGUAGE MODELS

We trained 4-gram LMs using the KenLM toolkit (Heafield, 2011), on Common Crawl data (Wenzek et al., 2020; Conneau et al., 2020), limiting the vocabulary to the most common 100k/200k words. For English, tri-grams and higher order grams occurring less than 3 times were pruned. Training data size (uncompressed), vocabulary size, LM vocabulary coverage of $D_U^{\text{tgt}}$ words, and LM perplexities are reported in Appendix, Table 8.

**LM Beam-Search Decoding** In all our experimental results, we report WER and CER, both with greedy and LM-beam search decoding. We rely on the lexicon-based beam-search decoder (with a word-based LM) from the `flashlight` framework (Kahn et al., 2022), ported in `torchaudio` (Yang et al., 2021). The same beam-search decoder is used to generate PLs in cross-lingual PL[5].

---

[3] `https://pypi.org/project/Unidecode`.

[5] We did not observe any significant impact of the unknown word score (set by default to $-\infty$) for PLs generation.

Table 3: Supervised monolingual acoustic models, for every language of interest, trained on Common Voice v12.0 data with a Transformer architecture (255M parameters). We report both character error (CER) and word error (WER) rates on validation ("Dev") and test ("Test") sets with greedy decoding ("greedy") and language model beam-search decoding ("w/ LM"). Beam size was set to 1k, and $\alpha, \beta$ hyper-parameters were tuned via random search.

| Lang. | # hours | greedy | | | | w/ LM | | | |
|---|---|---|---|---|---|---|---|---|---|
| | | Dev CER | Test CER | Dev WER | Test WER | Dev CER | Test CER | Dev WER | Test WER |
| en | 1552.8 | 4.9 | 6.9 | 14.6 | 17.8 | 4.7 | 6.2 | 11.0 | 13.0 |
| de | 801.2 | 1.6 | 2.2 | 6.8 | 7.9 | 2.1 | 2.5 | 9.2 | 9.7 |
| es | 395.6 | 1.8 | 2.1 | 6.7 | 7.4 | 1.9 | 2.1 | 5.8 | 6.2 |
| fr | 714.6 | 3.1 | 4.0 | 11.7 | 13.5 | 3.6 | 4.4 | 11.0 | 12.3 |
| rw | 1410.2 | 4.6 | 6.2 | 17.6 | 21.4 | - | - | - | - |
| sw | 48.6 | 4.3 | 5.9 | 15.1 | 18.9 | 4.1 | 5.6 | 11.6 | 14.9 |

## 5 RESULTS

### 5.1 ZERO-SHOT EVALUATION

In Figure 3, we report *source* language AMs performance (in WER/CER) on a *target* language in a zero-shot setting. While WER of zero-shot greedy decoding is around 90-100% (40-50% CER), WER of zero-shot beam-search decoding with a *target* language LM drops to 70-80% (30-40% CER) for some pairs of languages. English transfers the best to Spanish, German and Swahili, while every language transfers well to Spanish. Every *source* language performs poorly on French as a *target* language. Examples of transcriptions obtained from greedy and beam-search with an LM decoding of every *source* language AM applied to a *target* language are given in Appendix, Tables 9 and 10.

### 5.2 CROSS-LINGUAL PSEUDO-LABELING

*Target* AMs are initialized with *source* language AMs, before performing cross-lingual PL. PLs are generated with a beam-search decoding constrained with a *target* language LM: beam size is 100, $\alpha = 1, \beta = 0$ (with no hyper-parameters tuning). *Target* languages rw, ha, de, fr are decoded with 200k vocabulary size, while others with 100k. We perform IPL continuous AM training (as described in Section 3, Phase 1), updating the teacher which re-generates PLs after every 4k iterations: during first 4k iterations the *source* language AM acts as a teacher to bootstrap the model. From there, the teacher is updated with a new model's snapshot every 4k iterations. SpecAugment is activated after 1k iterations. The rest of the hyper-parameters are the same as discussed in Section 4.1 for monolingual baselines, except for the number of training iterations which is fixed to 50k iterations here. In Figure 3, we report WER and CER for final models with both greedy and LM beam-search decoding. The latter is tuned with a random search over $\alpha \in [0.3, 5], \beta \in [-10, 10]$, with the beam size set to 1000. As shown in Table 1 and in Appendix, Table 11, the cross-lingual PL constrained with a *target* LM does an impressive job at fixing PLs during the course of the *target* AM training.

**Indo-European Family** German and English are West Germanic languages with similar vocabularies, though their orthographies and phonetic inventories are very different (Wiese, 2000). French and Spanish are Romance languages, not Germanic, but like Germanic ones are in the Indo-European language family. In Figure 3, we observe that in general cross-lingual PL works great across Indo-European languages. Notably, any considered Indo-European language transfers well to Spanish (French to Spanish works best). With French as *target* language, cross-lingual PL performs poorly with any *source* language we tried[6].

**Niger-Congo Family** Kinyarwanda and Swahili are in the Niger-Congo language family, unrelated to the Indo-European ones, and written in the Latin alphabet, while Hausa is also an African language

---

[6]We explored variants, including (i) removing French accents by normalizing text to English tokens; (ii) increasing LM vocabulary to increase word coverage; (iii) character-based LM, as well as lexicon-free beam-search decoding for PL generation. None of these were able to improve results on French as a *target* language. We hypothesize that the main issue is related to the highly irregular orthography of French (Adda-Decker et al., 2005), which is too different from the other considered languages.

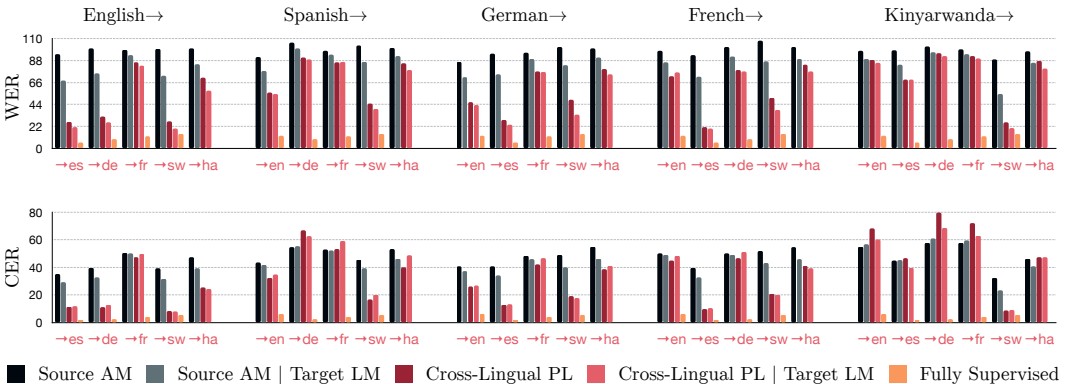

Figure 3: Zero-shot evaluation and cross-lingual pseudo-labeling word error (WER, top) and character error rates (CER, bottom) on Common Voice v12.0 for different *source* languages ($X \rightarrow$) with labeled data and *target* languages ($\rightarrow X$) with unpaired audio and text data: (i) zero-shot evaluation with a *source* acoustic model ("Source AM") on a *target* language; (ii) zero-shot evaluation of source AM coupled with a *target* language model ("Source AM | Target LM") via LM beam-search decoding (beam size is 100, $\alpha = 1, \beta = 0$, unknown words are not accepted); (iii) cross-lingual pseudo-labeling with greedy decoding ("Cross-Lingual PL") and LM beam-search decoding ("Cross-Lingual PL | Target LM"). Beam size is set to 1k and $\alpha, \beta$ are tuned via random search. Supervised models trained on the same *target* data, and decoded with LM beam-search are given as reference baselines.

Table 4: Impact of the Phase 2 with slimIPL on the final performance of cross-lingual pseudo-labeling.

| Language Pair | Greedy | | | | w/ LM | | | |
|---|---|---|---|---|---|---|---|---|
| | Dev CER | Test CER | Dev WER | Test WER | Dev CER | Test CER | Dev WER | Test WER |
| $en \rightarrow es$ (Phase 1) | 11.9 | 12.1 | 26.3 | 26.5 | 11.6 | 11.8 | 21.5 | 21.7 |
| $en \rightarrow es$ (Phase 2) | 9.9 | 10.0 | 23.2 | 23.5 | 10.0 | 10.1 | 19.1 | 19.3 |
| $en \rightarrow sw$ (Phase 1) | 7.3 | 8.4 | 24.5 | 27.1 | 6.6 | 8.0 | 17.8 | 20.0 |
| $en \rightarrow sw$ (Phase 2) | 6.1 | 7.4 | 20.9 | 23.7 | 5.6 | 6.8 | 15.7 | 18.0 |

but from the Afro-Asiatic family. Kinyarwanda transfers well to Swahili and poorly to Hausa. The latter is due to limited training hours (2.3h) in Hausa (see Section 5.4).

**Cross Family** While Kinyarwanda transfers poorly to any Indo-European language, any Indo-European language, surprisingly, transfers well to Swahili and somewhat well to Hausa (with the 2.3h training limitation for Hausa). In Appendix, Figure 9 we show that the quality of PLs improves as training goes, for Swahili as a *target* language. PL quality correlates with the AM WER (left). The LM has a critical impact on PL quality in the early stages of the training (right). An example of how PL transcriptions are evolving with training iterations is given in Appendix, Table 11.

### 5.3 BOOSTING WER PERFORMANCE WITH SLIMIPL

After Phase 1 (training with batch pseudo-labeling, e.g. IPL), we can improve performance in Phase 2 (online pseudo-labeling, using e.g. slimIPL), as described in Section 3. Table 5 shows the impact of Phase 2 using slimIPL on the $en \rightarrow es$ and $en \rightarrow sw$ language pairs. Consistent 1-4% absolute WER decreases are observed across greedy decoding and LM decoding for both pairs.

### 5.4 DATASET SIZE MATTERS

Perhaps unsurprisingly, WER performance of the *target* AM is correlated with the size of both the labeled audio dataset (for the *source* language), and the unlabeled audio dataset.

**Source Language: Labeled Audio** In Figure 4 (left), we show the importance of the amount of labeled audio available for the *source* language, for the $en \rightarrow sw$ pair (all available Swahili unlabeled data, 48.6h, is used). Subsets of different sizes were obtained by randomly sampling original training

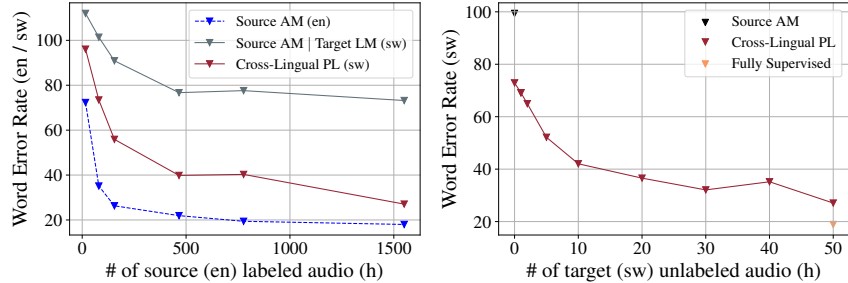

Figure 4: Cross-lingual PL dependence on the number of labeled data in the *source* language (left) and unlabeled audio in the *target* language (right) for $en \rightarrow sw$: for left we use all *target* language hours (50h) while for right we use all *source* language hours (1550h).

Table 5: Impact of the multilingual source AM on the final performance of cross-lingual PL.

| Language Pair | Greedy | | | | w/ LM | | | |
|---|---|---|---|---|---|---|---|---|
| | Dev CER | Test CER | Dev WER | Test WER | Dev CER | Test CER | Dev WER | Test WER |
| $en \rightarrow sw$ (zero-shot) | 37.7 | 39.3 | 98.6 | 99.4 | 29.6 | 31.8 | 70.2 | 72.9 |
| $(en, es, fr) \rightarrow sw$ (zero-shot) | 36.5 | 37.8 | 99.1 | 99.9 | 27.1 | 29.0 | 66.0 | 68.2 |
| $en \rightarrow sw$ (Phase 1) | 7.3 | 8.4 | 24.5 | 27.1 | 6.6 | 8.0 | 17.8 | 20.0 |
| $(en, es, fr) \rightarrow sw$ (Phase 1) | 6.6 | 7.9 | 20.4 | 23.2 | 6.7 | 8.1 | 16.9 | 19.5 |
| + 10x beam for PLs | 6.1 | 7.4 | 20.7 | 23.4 | 6.3 | 7.7 | 16.2 | 18.6 |
| $en \rightarrow sw$ (Phase 2) | 6.1 | 7.4 | 20.9 | 23.7 | 5.6 | 6.8 | 15.7 | 18.0 |
| $(en, es, fr) \rightarrow sw$ (Phase 2) | 6.0 | 7.2 | 19.9 | 22.6 | 5.7 | 6.9 | 15.5 | 17.6 |
| $en \rightarrow ha$ (zero-shot) | 34.0 | 47.3 | 95.1 | 100.0 | 31.0 | 39.6 | 88.8 | 84.4 |
| $(en, es, fr) \rightarrow ha$ (zero-shot) | 33.7 | 46.2 | 94.8 | 99.2 | 22.6 | 37.1 | 64.0 | 82.3 |
| $en \rightarrow ha$ (Phase 1) | 21.9 | 25.4 | 64.5 | 70.9 | 20.7 | 25.4 | 50.4 | 57.9 |
| $(en, es, fr) \rightarrow ha$ (Phase 1) | 19.4 | 23.5 | 62.1 | 67.8 | 19.0 | 23.2 | 46.2 | 54.4 |
| $(en, es, fr) \rightarrow ha$ (Phase 2) | 18.7 | 23.5 | 63.6 | 71.6 | 16.5 | 22.4 | 43.4 | 53.2 |

data, preserving the number of speakers, and proportion of hours per speaker. Any given set includes smaller ones. There is clear dependence on the performance of the source acoustic model: more labeled data available and better a source acoustic model the better cross-lingual transfer is performed.

**Target Language: Unlabeled Audio** In Figure 4 (right), we show the importance of the amount of unlabeled audio available for the *target* language, for the $en \rightarrow sw$ pair (all available English labeled data, 1552.8h, is used). Subsets of different sizes were obtained by randomly sampling original training data, preserving the number of speakers, and proportion of hours per speaker. Any given set includes smaller ones. While 1-2h is enough to significantly improve CER via cross-lingual PL, increasing unlabeled data size to 50h steadily improves model CER and WER performance.

## 5.5 MULTILINGUAL SOURCE AM

We ran a simple experiment using a single multilingual source AM to see if this would yield improvements for cross-lingual pseudo-labeling. We trained exactly the same model as all monolingual models (except we set dropout to a lower value of 0.05 to increase model capacity while all other hyper-parameters are exactly the same) on 3 languages combined together: English, Spanish and French (no balancing is done). The final performance (greedy decoding) of this multilingual model on validation sets is WER % [TER %]: on English 16.6 [5.4] (monolingual had 14.6 [4.9]); on Spanish 9.6 [2.3] (monolingual had 6.7 [1.8]); on French 13.8 [3.6] (monolingual had 11.7 [3.1]). Then we transfer it to Swahili and Hausa using exactly the same hyper-parameters as in the experiments when English monolingual model is transferred to them. The results below show that the multilingual AM consistently improves all results out of the box, even for Hausa with 2.3h only of training audio.

## 5.6 CROSS-LINGUAL PSEUDO-LABELING ACROSS ALPHABETS

The core of this paper reports results with languages from the same alphabet (Latin). We show here that unsupervised cross-lingual PL can be easily extended to the case where *source* and *target*

Table 6: Supervised baselines and unsupervised ASR via cross-lingual pseudo-labeling from Belarusian (be, Cyrillic alphabet, 470h) to Czech (cs, Latin alphabet, 25h).

| Language | Greedy | | | | w/ LM | | | |
|---|---|---|---|---|---|---|---|---|
| | Dev CER | Test CER | Dev WER | Test WER | Dev CER | Test CER | Dev WER | Test WER |
| *be* (470h) *(original token set)* | 0.8 | 0.8 | 4.1 | 4.3 | 1.4 | 1.5 | 6.4 | 6.6 |
| *be* (470h) *(modified token set)* | 0.9 | 0.9 | 4.4 | 4.5 | 1.4 | 1.4 | 6.4 | 6.4 |
| *cs* (25h) | 7.9 | 8.7 | 30.6 | 32.5 | 6.8 | 7.5 | 19.9 | 21.0 |
| *be* → *cs* (zero-shot) | 45.9 | 45.7 | 100.8 | 101.1 | 41.3 | 40.8 | 80.7 | 79.6 |
| *be* → *cs* (Phase 1) | 26.3 | 26.6 | 61.9 | 62.6 | 26.8 | 27.0 | 57.9 | 58.3 |

Table 7: Zero-shot ("zero") and cross-lingual pseudo-labeling ("pl") on LJSpeech test set, with Common Voice source AMs: CER (left) and WER (right) with greedy ("greedy") and an LM beam-search ("beam") decoding. We compare with character-based wav2vec-U 2.0 Liu et al. (2022).

| Source | CER | | | | WER | | | |
|---|---|---|---|---|---|---|---|---|
| | zero-greedy | zero-beam | pl-greedy | pl-beam | zero-greedy | zero-beam | pl-greedy | pl-beam |
| *fr* → | 48.8 | 48.6 | 44.9 | 47.5 | 96.8 | 87.2 | 81.9 | 76.4 |
| *es* → | 47.6 | 48.3 | 41.7 | 42.1 | 96.1 | 85.7 | 78.0 | 75.1 |
| *de* → | 35.8 | 32.6 | 25.8 | **27.3** | 85.5 | 66.2 | 55.6 | **48.3** |
| wav2vec-U 2.0 | - | - | - | 34.6 | - | - | - | 64.0 |

languages have different alphabets. We have shown in Section 5.4 (Figure 4) that the size of *source* language labeled data is critical for the cross-lingual PL approach to work. In Figure 3, we also showed that language proximity matters. As Common Voice v12.0 is rather biased towards Latin languages (in terms of available data), we ended-up picking Belarusian (be), a Cyrillic-based language, as *source* (with 470h of labeled audio), and chose Czech (cs) *target* language (rather low resource in this dataset, with only 25h hours of data). Belarusian and Czech languages share pronunciation similarities and similar sounds like 'ч' and 'č'.

To perform cross-alphabet transfer, we followed basic character-based transliteration rules from Belarusian Cyrillic alphabet to Latin alphabet[7]. These rules cover some specific sound similarities in both languages, e.g. 'ч' and 'č', 'ш' and 'š', see Appendix, Table 12. This allows us to train a Belarusian *source* AM, with tokens ("modified token set") matching Latin tokens. The cross-lingual PL procedure can then be performed normally. We show WER performance in Table 6.

### 5.7 COMPARISON TO WAV2VEC-U 2.0 ON LJSPEECH

We report performance on the LJSpeech dataset in Table 7, following wav2vec-U 2.0 (Liu et al., 2022) for the setup. Train/dev/test (22.8h/0.6h/0.6h) splits and transcription normalization (lower casing and punctuation removal) are identical. For zero-shot evaluation on LJSpeech we use *source* language AMs trained on Common Voice (Table 3), and a 4-gram LM trained on LibriSpeech LM corpus with 200k vocabulary (perplexity of 148 on *dev-clean* and 137 on *dev-other*). Decoding hyper-parameters were not tuned (beam size of 100 and $\alpha = 1, \beta = 0$). Cross-lingual PL is done in the same way as in Section 5.2. We outperform wav2vec-U 2.0 by 15% absolute WER, using 800h of labeled German data only, instead of 60k hours of unlabeled English data.

## 6 CONCLUSIONS

We demonstrated in this work that unsupervised learning via cross-lingual pseudo-labeling can be very effective. Our method is significantly simpler than existing unsupervised speech recognition methods, relying only on standard semi-supervised learning recipes and dispensing with GANs, phoneme labels, and random silence insertion. Reasonable acoustic models for several *target* languages were trained, using no labeled audio data from these *target* languages. This opens new avenues to low-resource languages, e.g. as we achieve word error rate of 23.7% (18%) with greedy (beam-search) decoding, for cross-lingual pseudo-labeling from Indo-European English to Niger-Congo Swahili languages. Future work will include more language pairs, and advance multilingual source acoustic models.

---

[7]https://en.wikipedia.org/wiki/Belarusian_Latin_alphabet

## ETHICS STATEMENT

In the paper, we aim at understanding connection between automatic speech recognition for different languages and how to perform transfer from some source high resource languages to target low resource languages without supervision on the target side. We hope this is a positive contribution towards under-represented data sources for ASR. While one can imagine ASR being used for negative purposes, it is our hope that the advantages generated by improving ASR for low-resource settings outweigh its possible negative uses.

## REPRODUCIBILITY STATEMENT

For all experiments we use publicly available datasets for research: LibriSpeech/LJSpeech (CC BY 4.0/Public Domain) and Common Voice v12.0 (CC BY-SA 3.0). Data processing is described in the main body of the paper. We tried as much as possible to describe all results, configurations, training details, and hyper-parameters throughout the paper and in Appendix.

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

## A  DISCUSSION AND LIMITATIONS

Our proposed unsupervised ASR method via cross-lingual pseudo-labeling aims to simplify the huge machinery of existing methods for unsupervised ASR. In addition to experimenting with Germanic, Romance, Slavic, Niger-Congo, and Afro-Asiatic languages, we also showed that our method can be applied to languages with very different orthographies and different alphabets. However, for successful cross-lingual pseudo-labeling there should be enough unlabeled data in the *target* language ($\geq$ 30h of audio-only), and the *source* model should have reasonable performance (e.g. trained on 100-500h of paired audio-text data) as well. We also highlight, that *source* and *target* languages should share phonetic similarities, e.g. we were not able to perform cross-lingual pseudo-labeling on French language as a *target*.

In the paper, we restrict our comparison only to the character-based unsupervised methods as any phoneme-based system will be better due to a very strong inductive bias. However, if a language does not have labeled audio data, it is even less likely that it will have a phoneme-based lexicon, making our method more applicable to real-world scenarios (if phonemes are available, it makes more sense to use a method like ASR2K (Li et al., 2022)).

# B    DETAILED RESULTS ON COMMON VOICE v12.0

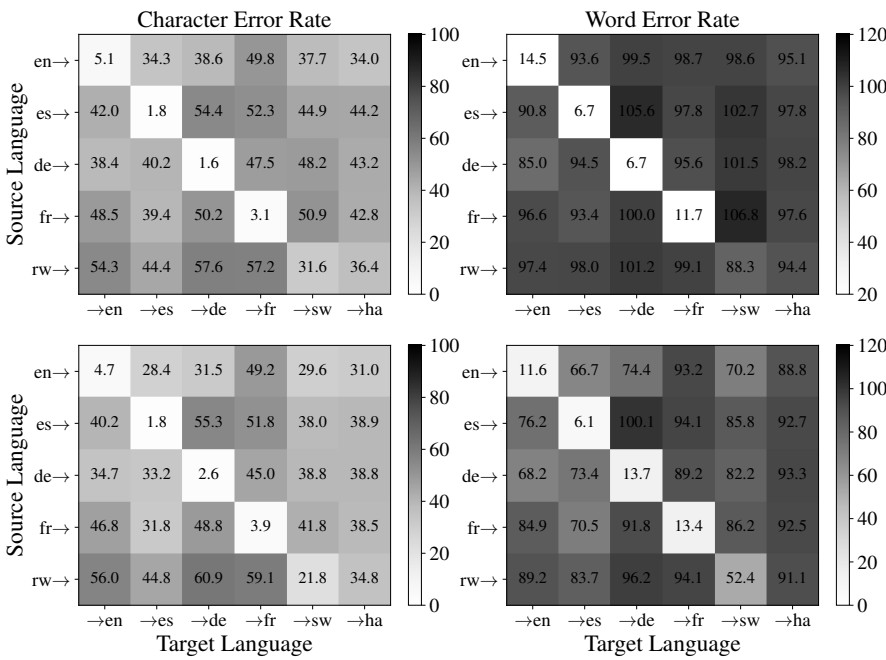

Figure 5: Zero-shot CER (left) and WER (right) with greedy (top) and LM beam-search decoding (bottom) on Common Voice **validation sets**, for models trained on a *source* language $X \rightarrow$ and transferred to a *target* language $\rightarrow X$. Beam size is set to 100 and $\alpha = 1, \beta = 0$. We found that German LM decoding is worse than greedy decoding because unknown words are not accepted in the decoding process.

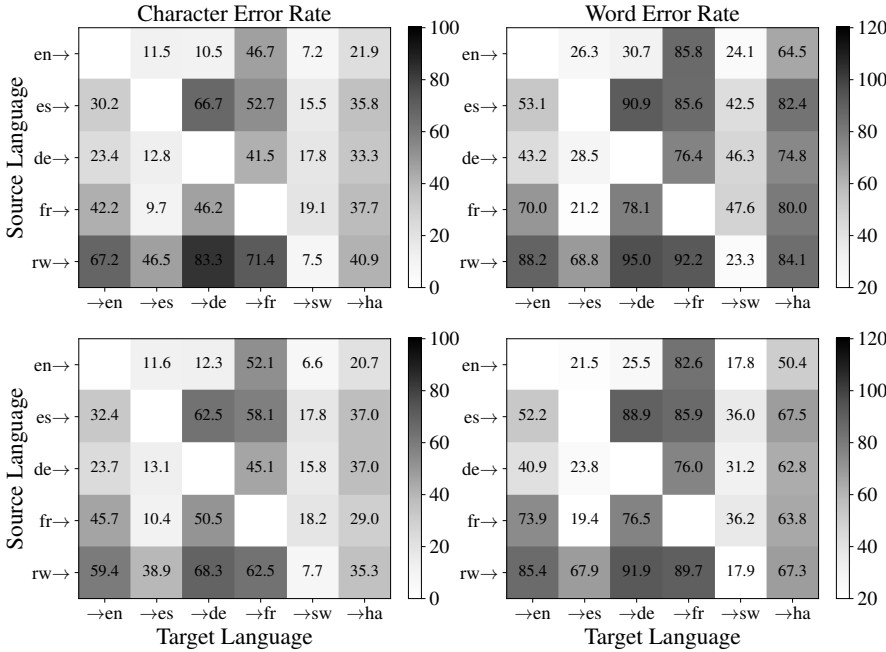

Figure 6: Cross-lingual PL (from a *source* language $X \rightarrow$ to a *target* language $\rightarrow X$) CER (left) and WER (right) with greedy (top) and LM beam-search decoding (bottom) on Common Voice **validation sets**. Beam size is set to 100 and $\alpha = 1, \beta = 0$ during cross-lingual PL (top). Beam size is set to 1k and $\alpha, \beta$ are tuned via random search for the LM beam-search decoding results (bottom).

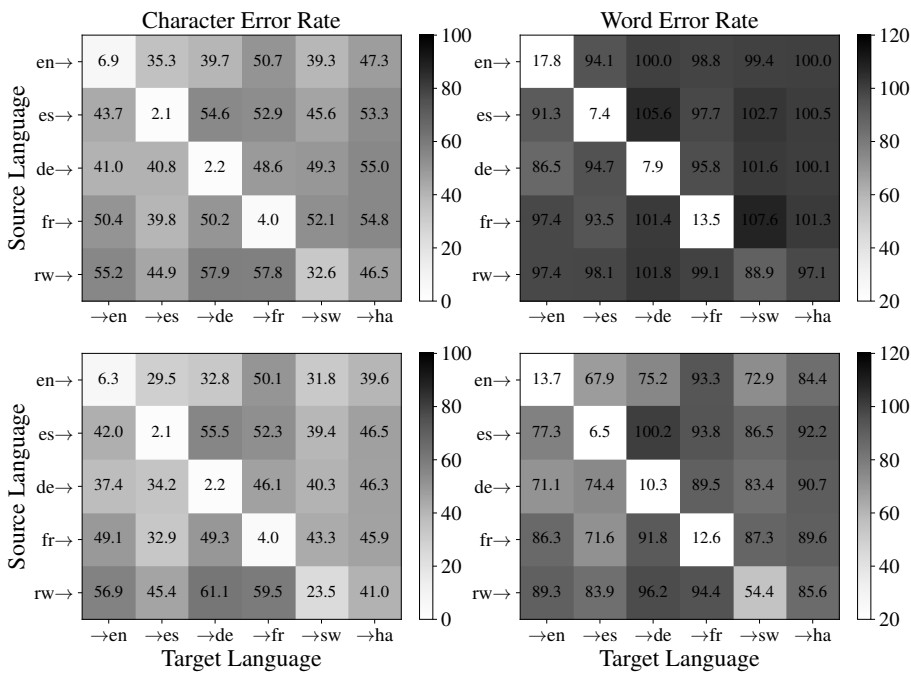

Figure 7: Zero-shot CER (left) and WER (right) with greedy (top) and LM beam-search decoding (bottom) on Common Voice **test sets** for models trained on a *source* language $X \rightarrow$ and transferred to a *target* language $\rightarrow X$. Beam size is set to 100 and $\alpha = 1, \beta = 0$. Unknown words are not accepted, so German LM language decoding is worse than greedy one.

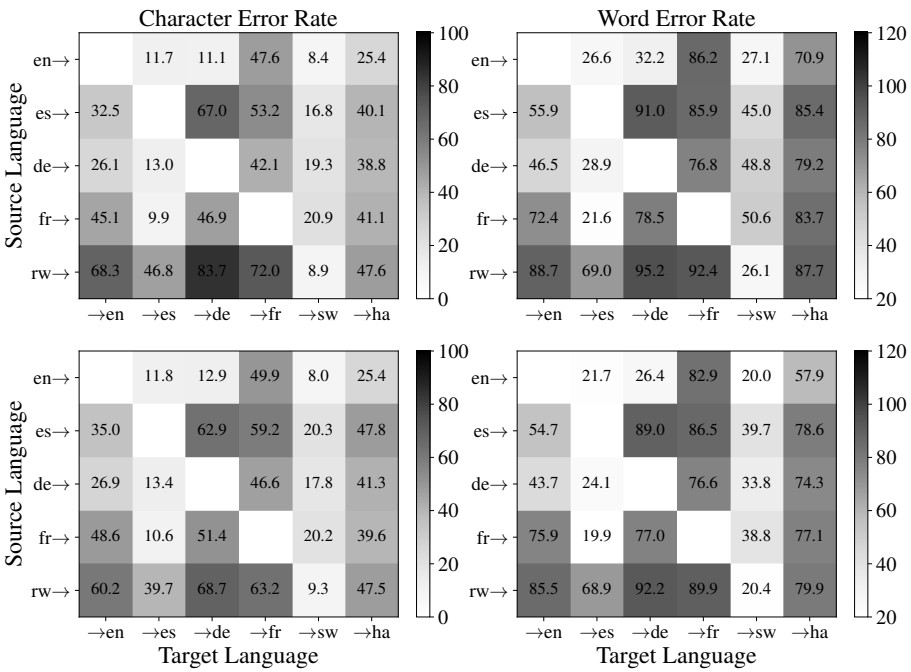

Figure 8: Cross-lingual PL (from a *source* language $X \rightarrow$ to a *target* language $\rightarrow X$) CER (left) and WER (right) with greedy (top) and LM beam-search decoding (bottom) on Common Voice **test sets**. Beam size is set to 100 and $\alpha = 1, \beta = 0$ during cross-lingual PL (top). Beam size is set to 1k and $\alpha, \beta$ are tuned via random search for the LM beam-search decoding results (bottom).

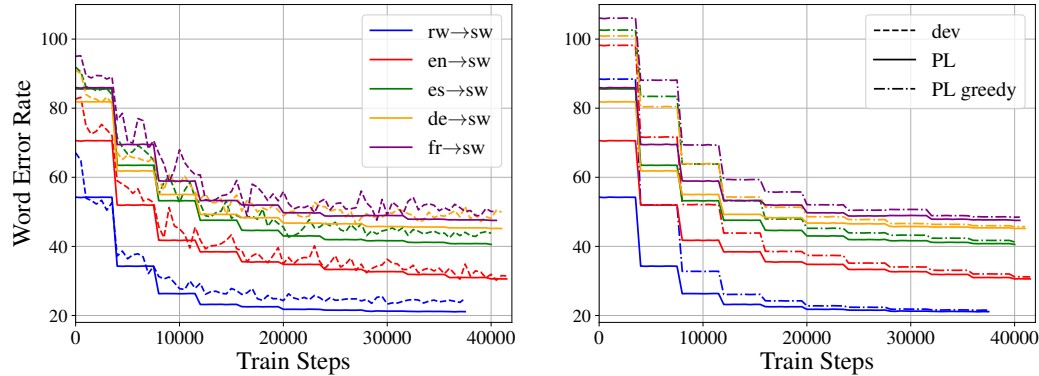

Figure 9: PLs quality on Swahili as a *target* language, for different *source* languages. Solid lines are actual PL WER, obtained via LM beam-search decoding. Impact on the dev set WER (dashed lines) is shown on the left. Greedy (dotted-dashed) and LM-beam search decoding PLs are compared on the right.

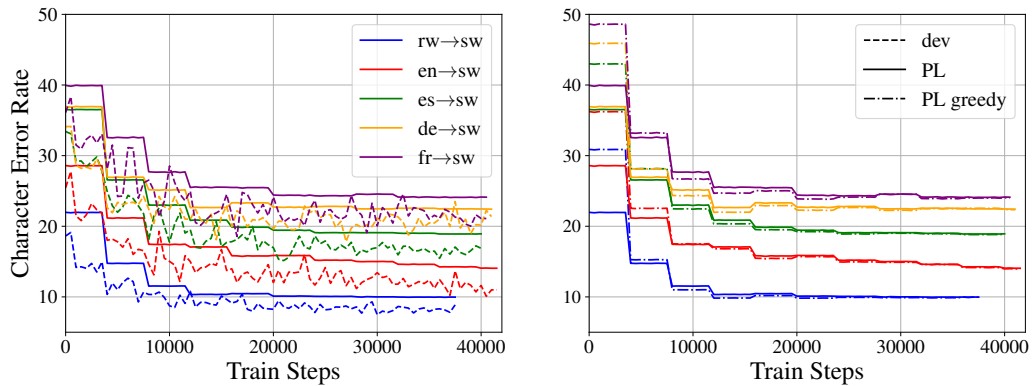

Figure 10: PLs quality on Swahili as a *target* language, for different *source* languages. In Figure 9 we reported WER, here we report CER. Solid lines are PL CER, obtained via LM beam-search decoding. Impact on the validation set CER (dashed lines) is shown on the left. Greedy (dotted-dashed) and LM beam-search decoding PLs are compared on the right.

Table 8: Perplexity (PPL) of 4-gram LMs for the target languages trained on Common Crawl data and evaluated on Common Voice dev and test sets. Perplexity is reported both with and without out-of-vocabulary (OOV) words. Vocabulary sizes and LM vocabulary coverage of $D_U^{\text{tgt}}$ words in train data are also provided.

| Lang. | Data (GB) | Model (GB) | Vocab. | Vocab. $D_U^{tgt}$ | Coverage (%) | w/o OOV PPL Dev | Test | w/ OOV PPL Dev | Test |
|---|---|---|---|---|---|---|---|---|---|
| en | 288 | 28 | 100k | 290k | 96.5 | 171 | 150 | 230 | 190 |
| de | 64 | 49 | 200k | 250k | 95.6 | 228 | 228 | 350 | 345 |
| es | 52 | 32 | 100k | 92k | 97.7 | 141 | 140 | 180 | 177 |
| fr | 54 | 37 | 200k | 240k | 96.2 | 168 | 164 | 254 | 250 |
| sw | 1.6 | 1.5 | 200k | 194k | 97.4 | 175 | 194 | 235 | 270 |
| ha | 0.3 | 0.4 | 200k | 16k | 99.0 | 295 | 307 | 316 | 339 |

## C   TRAINING DETAILS ON PHASE 2 WITH SLIMIPL

Phase 1 training (see Section 3) bootstraps an AM with Iterative Pseudo-Labeling. We found we can improve WER performance via a Phase 2, performing online pseudo-labeling with slimIPL. We

Table 9: Example of pseudo-labels generated by source AMs on English, German and Spanish audios with greedy (top) or beam-search (bottom) decoding on Common Voice.

| AM | English
but what is the use of talking | German
da schließen wir die werften | Spanish
al surcon la república del perú |
|---|---|---|---|
| **en** | but what is thee use of talking
but what is the use of talking | dashlees in veerti we aften
das les in vierte werften | alsur conorapulita alpero
al sur con apulia albero |
| **de** | der ort ist sie istaki
der ort ist sie stake | da schließen wir die werften
da schließen wir die werften | also colarebulidal bero
alto color pulida pero |
| **es** | thaweseius tarke
the wastes take | tashlis teuti deav
das listet das | al sur con la república del perú
al sur con la república del perú |
| **fr** | pourquoi si lise craque
pour quoi cilic croque | tashlisn vit à diveraf
das les via divers | alsoud qon la lacoulit à e le pelo
al sur con la la colita del pelo |
| **rw** | aa isi yise trak
isi is track | yasheyizindi ya arire hafte
das india arie hafte | azur conal republika ya eberu
azur con republica peru |

Table 10: Example of pseudo-labels generated by source AMs on French, and Swahili audios with greedy (top) or beam-search (bottom) decoding on Common Voice.

| AM | French
cet écureuil volant est endémique d'indonésie | Swahili
kamwe vilio havizuii jambo kutokea kama limepangwa |
|---|---|---|
| **en** | setekuri volom etonde mik dandonizi
sete kiri volume onde mick dando | kam me vileo havizui jamba kutaker kamali mepanga
kamwe vilio havijui jambo kutaka kama imepanga |
| **de** | ceticurivola etandemique den donisi
cette cure vola et an demie de denise | caumo vilio havisouidiambrotea kamer in ne bamburg
kama vilio havizidi boti kama in bambo |
| **es** | seteque hy vollow están de mik dando nesi
sete que y vallon stan de mick danton si | camo y filio a bisui lamboco thaqueo camarmi fangua
kama vilio visu jambo co cafe camara panga |
| **fr** | cet écureuil volant est endémique d'indonésie
cet écureuil volant est endémique d'indonésie | camovili aura vésoui bien beaucoup de rir comme elle n'est pas loi
kama vile aura visu bien eacop dera come elle nest pas moi |
| **rw** | setecuovi volom et ondemiqe dendonesi
cette cuve villon et de mike donde nes | kanwa vilio habizuiriyambokotsako ya kamarine pangwa
kama vilio habibu ambako taka ya kamari ni pangwa |

Table 11: How PLs evolve with training iterations in cross-lingual pseudo-labeling, with $en \rightarrow sw$: (top) golden and (bottom) PLs.

| iteration | **hapa ni mahali ambapo wazee wetu walipatumia kama darubini** |
|---|---|
| 0-4k | hapani mali ambapo was a watu alipotumia kama darubini |
| 4k-8k | hapa ni mali ambapo was watu alipotumia kama darubini |
| 8k-12k | hapa ni mahali ambapo wa watu walipotumia kama darubini |
| 12k-16k | hapa ni mali ambapo wawatu walipotumia kama darubini |
| 16k-20k | hapa ni mahali ambapo wawatu walipotumia kama darubini |

report here technical details to reproduce these experiments. First, we take a *target* AM obtained by Phase 1, and generate PLs using a LM beam-search decoding process, with a beam size larger than the one used in Phase 1 training (set to 1000). Decoding hyper-parameters $\alpha$ and $\beta$ are tuned on the validation set. PLs are then obtained for the whole train set of the *target* language. We then train a new *target* AM, initializing it from the *source* AM, and fine-tuning it with the PLs obtained as mentioned above. Fine-tuning is done with the same training configuration as listed in Section 4.1, except for Swahili (where we fine-tune for 15k iterations) and Spanish (fine-tuned for 20k iterations) as *target* languages. After this, we continue the training with slimIPL (Likhomanenko et al., 2021) (the optimizer state was not reset). The slimIPL PLs cache size is set to 100, and cache probability is 0.1 up to additional 30k steps. Compared to published slimIPL settings, all data is here unlabeled, so the proportion of labeled data is set to 0. We found that re-initializing the AM from the *source* AM in the Phase 2 is important to further boost the WER performance.

Table 12: Mapping between Belarusian Cyrillic alphabet and Czech Latin alphabet.

| Belarusian | Czech |
| --- | --- |
| а | a |
| б | b |
| в | v |
| г | h |
| ґ | g |
| д | d |
| дь | ď |
| е | e |
| ё | ě |
| ж | ž |
| з | z |
| зь | ź |
| і | i |
| й | j |
| к | k |
| кв | q |
| кс | x |
| л | l |
| м | m |
| н | n |
| нь | ň |
| о | o |
| п | p |
| р | r |
| рж | ř |
| с | s |
| сь | ś |
| т | t |
| ть | ť |
| у | u |
| ў | ŭ |
| ф | f |
| х | ch |
|ць | ć |
| ц | c |
| ч | č |
| ш | š |
| ы | y |
| э | é |
| ю | ju |
| я | ja |
| ль | í |

