# OpenReview forum: "Unsupervised ASR via Cross-Lingual Pseudo-Labeling"
_ICLR.cc/2024/Conference — Submitted to ICLR 2024_

### Official Review · Reviewer_GYQT · 2023-10-28

**Soundness:** 3 good
**Presentation:** 3 good
**Contribution:** 2 fair
**Rating:** 6
**Confidence:** 4

**Summary:**

This paper introduces an innovative approach for training unsupervised Automatic Speech Recognition (ASR) systems in low-resource languages, using only unmatched speech and text data. The method takes advantage of the similarity in pronunciation between characters in a paired language to create pseudo-labels for the target language, which are further refined by a language model. This approach demonstrates promising results in both cross-language and cross-family scenarios.

**Strengths:**

1. This paper addresses a practical and underexplored problem. Implementing the proposed method in the pseudo-labeling framework is more straightforward than using GAN-based approaches.
2. This approach demonstrates promising results in both cross-language and cross-family scenarios.
3. The performance is further enhanced with the use of multi-lingual source AMs.
4. The paper is well-written and straightforward to understand.

**Weaknesses:**

1. The results are not particularly encouraging:
    * It would be more valuable to demonstrate that a single source AM can be effectively applied to multiple target languages (e.g., en -> {sw, ha, X, Y, Z}) rather than showcasing the application of multiple source AMs to the same target language (e.g., {en, es, ge, fr} -> sw):
    * It appears that this method is effective primarily for the language pairs { *-> sw} and {be -> cs}, which raises concerns about its generalization to other low-resource languages.

2. In contrast to wav2vec-u, the concept of "similarity" might restrict the applicability of this method, potentially preventing it from identifying an appropriate source language for a specific target language.

**Questions:**

1. In the paper, it is claimed that "for African languages, we use Kinyarwanda only as a source language (as text data are not available in the Common Crawl dataset)." How is the Kinyarwanda AM trained without text data?
2. Table 3 indicates that the performance of the de degrades when language model decoding is applied. Is there any analysis provided in the paper to explain this

---

> ### Author Response · Authors · 2023-11-14
> **Authours Rebuttal**
>
> We would like to thank Reviewer GYQT for their time. Please find below our comments:
>
> > It would be more valuable to demonstrate that a single source AM can be effectively applied to multiple target languages (e.g., en -> {sw, ha, X, Y, Z}) rather than showcasing the application of multiple source AMs to the same target language (e.g., {en, es, ge, fr} → sw):
>
> Thanks for the interesting suggestion! We will investigate this in the future.
>
> > It appears that this method is effective primarily for the language pairs { *-> sw} and {be -> cs}, which raises concerns about its generalization to other low-resource languages.
>
> We wish we could have included more low-resource languages settings. However, the main obstacle is the availability of the unlabeled audio data and unpaired text for the same language. Apart Common Voice, we are not aware of many freely available datasets with 30h or more unlabeled audio per language, and large available text corpora.
>
> > In contrast to wav2vec-u, the concept of "similarity" might restrict the applicability of this method, potentially preventing it from identifying an appropriate source language for a specific target language.
>
> We agree. wav2vec-U 2.0 and our approach have different tradeoffs. We trade (phoneme lexicon, GAN training, and large amount of unlabeled target audio) for (simple pseudo-labeling, limited amount of labeled source audio). In practice, as labeled source audio is always available, our approach is most likely applicable. In contrast, wav2vec-U 2.0 requires a phoneme lexicon and large amount of target audio, which may not always be available. As for which source language to choose, zero-shot performance on target language seems to be a good indicator of the cross-lingual transfer difficulty (see Figure 3).
>
> > In the paper, it is claimed that "for African languages, we use Kinyarwanda only as a source language (as text data are not available in the Common Crawl dataset)." How is the Kinyarwanda AM trained without text data?
>
> Kinyarwanda has labeled audio data (with transcriptions) in Common Voice, which is used to train the source acoustic model for Kinyarwanda. However, no available unpaired text data to train a language model is available. If we had used the audio transcriptions for that matter, then audio and language model would have the same training text data, in which case the problem could be viewed as a simple alignment problem rather than unsupervised ASR.
>
> > Table 3 indicates that the performance of the de degrades when language model decoding is applied. Is there any analysis provided in the paper to explain this
>
> Thanks for the question! As German uses compound words, the language model dictionary has limited word coverage. This shows at beam-search decoding time, when unknown words (not in the dictionary) are then decoded into multiple words allowed by the dictionary. We found that if we allow the beam-search decoding to produce unknown words, then it mitigates the issue, at the expense of an additional hyper-parameter. For simplicity, and consistency with other languages which did not require it, we sticked to beam-search decoding restricted to the original dictionary.

---

> > ### Comment · Reviewer_GYQT · 2023-11-23
> >
> > I appreciate the author's response. The only thing I would like to confirm from the author is that for all target languages, there's no overlap between audio transcripts and LM training data (Common Crawl).  If this condition is met, I would prefer to retain the current score.

---

> > > ### Author Response · Authors · 2023-11-23
> > > **Additional comments**
> > >
> > > Dear Reviewer GYQT,
> > >
> > > Thanks for your reply and important question! Based on the papers cited in https://data.statmt.org/cc-100/, authors of cc-100 built it without Wikipedia data but by finding content close to Wikipedia. The latter is done by evaluating LM (trained on the Wikipedia) on the common crawl data. Based on the description in the Common Voice (CV) dataset paper, particularly protocol on adding a new dataset in Section 4.2, it is mainly Wikipedia sentences that used to record the audio (though extra sentences can be also added). Thus, intersection between the datasets is unlikely.
> > >
> > > For certainty, we evaluated the percentage of CV audio samples (their transcription) that appear in the LM text corpus and the percentage of samples in LM text corpus that appear in the CV for all target languages, shown in Table below. Thus, a data leak in LM used for the cross-lingual PL is unlikely.
> > >
> > > | Target language | Amount of audio samples that appear in the text data (%) | Amount of text samples that appear in the audio data (%) |
> > > | ---| ---| ---|
> > > | sw | 1.16 | 0.015 |
> > > | ha | 0.05 | 3e-5|
> > > | cs | 1.02 | 0.07|
> > > | es | 1.21 | 1.6e-3 |
> > > | de | 0.93 | 2.5e-3 |
> > > | fr | 1.23 | 1.8e-3 |
> > > | en | 1.32 | 1.1e-3 |

---

### Official Review · Reviewer_mAuZ · 2023-10-31

**Soundness:** 3 good
**Presentation:** 3 good
**Contribution:** 3 good
**Rating:** 6
**Confidence:** 5

**Summary:**

This paper describes an approach to training an ASR model without transcribed speech in a language – “Unsupervised ASR”.  The central idea is to use an ASR model for some high resource source language to generate hypotheses for the target language, and integrate target language information through an LM. Then iterative pseudolabeling can be used to refine performance.  The authors show competitive performance.  The authors also demonstrate performance if the source and target language do not share a writing script.

**Strengths:**

The clearest strength of this approach is its simplicity.  The technical approach uses mostly off-the-shelf, well understood techniques to solve a challenging task.

Evaluations are largely well done across a variety of language families (though more limited than other work on unsupervised ASR)

**Weaknesses:**

It would be good to compare performance on more languages, prior work has investigated FLEURS.  Comparing performance on this data set would enable clearer comparisons.

There is a reliance on the unidecode to “romanize” the character sets of various languages to Latin script.  The recognition performance in a language should be in its own script, not a romanized version.  It does not seem as though unidecode inverted prior to CER calculation. (Though I suppose this could also be a “question” rather than a “weakness”

**Questions:**

Is the example in Figure 1 a real example or made up to demonstrate idealized behavior? This wasn’t clear from the context.

In Section 2.0 it is claimed that ‘unsupervised ASR is viable, as long as source and target language share enough ”similarities”’.  Which similarities are critical for performance here? Acoustic, lexical, other?

Just a note for Section 5.6 – the title describes transfer across “Alphabets”, however, not all writing systems are alphabets. Would this approach extend to abugida, abjad or logographic writing systems?

Note: it would be nice if Figure 4 used the same Y axis in both tables.  (It’s more understandable that the X axis varies)

---

> ### Author Response · Authors · 2023-11-14
> **Authours Rebuttal**
>
> We would like to thank Reviewer mAuZ for their time. Please find below our comments:
>
> > It would be good to compare performance on more languages, prior work has investigated FLEURS. Comparing performance on this data set would enable clearer comparisons.
>
> As prior work on unsupervised ASR did not report numbers on FLEURS, we did not consider it, as we would not be able to compare with them. Also, as far as we know FLEURS is used to compare multilingual systems trained on data of internet-scale, which is not the type of system trained in this paper.
>
> > There is a reliance on the unidecode to “romanize” the character sets of various languages to Latin script. The recognition performance in a language should be in its own script, not a romanized version. It does not seem as though unidecode inverted prior to CER calculation. (Though I suppose this could also be a “question” rather than a “weakness”
>
> We used unidecode only to “romanize” characters which are not in the native character set of each of the language considered. In practice, we found this occurs rarely (< 0.01%). Note that Common Voice datasets come unnormalized (for example, they contain punctuation), and using unidecode is a way to normalize the Common Voice datasets in a generalizable manner (across different languages), with little effect on the WER (given those events occur rarely). We will release the normalization procedure for reproducibility.
>
> > Is the example in Figure 1 a real example or made up to demonstrate idealized behavior? This wasn’t clear from the context.
>
> It is a real example from the es → en model. We will clarify it.
>
> > In Section 2.0 it is claimed that ‘unsupervised ASR is viable, as long as source and target language share enough ”similarities”’. Which similarities are critical for performance here? Acoustic, lexical, other?
>
> Based on our empirical results in Figure 3, languages which are known to have acoustic similarity transfer better. In addition, if the target language has simple pronunciation rules (like Spanish), transfer works better in practice. In contrast, French (known for its highly irregular orthography (Adda-Decker et al., 2005)) is a difficult target language, as inferring those rules is hard.
>
> Zero-shot performance on target language seems to be a good indicator of the cross-lingual transfer difficulty (see Figure 3).
>
> > Just a note for Section 5.6 – the title describes transfer across “Alphabets”, however, not all writing systems are alphabets. Would this approach extend to abugida, abjad or logographic writing systems?
>
> Using an adequate transliteration system, one could convert logographic writing systems to other alphabets, and still use our approach. For example, converting Chinese hanzi → pinyin (e.g. ”你好“ → ”ni hao“) or Japanese kanji → romaji (e.g. ”日本“ → ”Nihon“). As the point of the paper is to show how and when end-to-end unsupervised ASR via cross-lingual PL can work, we did not investigate cases where complex transliteration systems are required though. We will change ”across alphabets“ in the section title to ”across writing systems“.
>
> > Note: it would be nice if Figure 4 used the same Y axis in both tables. (It’s more understandable that the X axis varies)
>
> Thanks for the suggestion! We will fix it.

---

### Official Review · Reviewer_26fF · 2023-11-03

**Soundness:** 3 good
**Presentation:** 3 good
**Contribution:** 2 fair
**Rating:** 3
**Confidence:** 5

**Summary:**

This paper tackles multilingual ASR, especially for dealing with unsupervised training in target languages given source languages' ASR systems trained on supervised source data. The idea is simple but powerful. The method uses PL techniques but extends it for a transfer learning scenario from source to target languages. The paper investigates this transfer learning capability with various dimensions (e.g., across the language, across the language family, variants of PL methods, amount of labeled source data and unlabelled target data, the use of LMs, etc.). With these investigations, the method finally achieved sufficient performance in some language pairs. Although most languages are based on the standard Latin scripts, the paper also shows the potential of applying this method to target languages with unseen scripts with the help of a transliteration technique.

**Strengths:**

- Multilingual ASR is an important research topic to bridge the digital divide in underrepresented regions.
- The proposed method does not require labeled data for the target languages
- Intensive analyses of the proposed methods with various dimensions

**Weaknesses:**

- Although the topic is very important, the technique itself does not have sufficient novelty as an ML conference paper. The topic is specific to ASR, and the technique is based on one particular ASR method (i.e., CTC). The connection to general ML problems is not clear.
  - for example, some experimental results (e.g., the use of n-gram LMs, etc.) are specific to CTC, and it does not seem to be generalized to the other architectures.
- Most experimental findings are expected, and there are not so many new findings (e.g., it's a bit trivial that large data help the performance, etc.).
- The methodology is not very new. Although there are several differences, some prior studies try to transcribe unseen languages with seen language ASR systems (e.g., Hasegawa-Johnson, Mark A., et al. "ASR for under-resourced languages from probabilistic transcription." IEEE/ACM Transactions on Audio, Speech, and Language Processing 25.1 (2016): 50-63.).

**Questions:**

- Did you use SSLs? Since SSLs are obtained by unsupervised training, combining this method and SSL would be very powerful.
- Can you explain why this method uses CTC? Is there a particular reason, or is this method applicable to the other methods (e.g., HMM, attention-based encoder-decoder, RNN-T)? I think that this part makes the paper's scope narrow.
- Section 4.1 "(iii) converting characters into the Latin token set via unidecode3 package; characters failing the conversion were discarded from the text": Can you describe the examples? Also, I'm concerned that if we discard some characters in the reference, we cannot evaluate the performance validly with the other reports. Can you clarify this part?
- Section 5.7: It is difficult to conclude since they are very different to compare (e.g., I'm not sure which one is more difficult using 800h of labeled German data and 60k hours of unlabeled English data). Can you explain the benefits of your method more clearly?

Other suggestions
- For me, the method is a little bit over-claimed since this method is primarily applicable to languages with the same scripts or with transliteration systems. It would not be easy to apply this method to the ideogram languages (e.g., Chinese and Japanese, although they are rich-resource languages, and we can build ASR systems easily). I think the paper requires some discussion about it (e.g., adding it to the DISCUSSION AND LIMITATIONS section?).
- Section 2.2: $(\alpha > 0)$ suddenly appears. This should be shown around Eq. (1), where $\alpha$ first appears.
- Section 4, first paragraph: These experimental setups are difficult to follow, as they are very diverse. I recommend you describe the design of the experiment (or the intention of what you want to show) when you describe each experimental setup.
- Figure 3 is too small... Please improve it.

---

> ### Author Response · Authors · 2023-11-14
> **Authours Rebuttal**
>
> We would like to thank Reviewer 26fF for their time. Please find below our comments:
>
> > Although the topic is very important, the technique itself does not have sufficient novelty as an ML conference paper. The topic is specific to ASR, and the technique is based on one particular ASR method (i.e., CTC). The connection to general ML problems is not clear. for example, some experimental results (e.g., the use of n-gram LMs, etc.) are specific to CTC, and it does not seem to be generalized to the other architectures.
>
> Unsupervised **end-to-end** ASR that practically works is a significant novelty brought by this paper. End-to-end ASR alleviates the need for any phoneme lexicon required in other successful unsupervised ASR approaches (like wav2vec-U 2.0). Another notable difference with wav2vec-U 2.0 is that we do not rely on adversarial training at all. We will highlight better these two points in the paper.
>
> We truly think that speech-specific papers have their place at ICLR. Prominent work in ASR such as wav2vec 2.0 or wav2vec-U were published at NeurIPS and Whisper was published at ICML.
>
> Our method is generalizable to any acoustic or language models which can output posterior probabilities (see Eq. 1). There is nothing specific in our approach which limits ourselves to CTC (or Transformers acoustic models).
>
> > Most experimental findings are expected, and there are not so many new findings (e.g., it's a bit trivial that large data help the performance, etc.).
>
> As stated above, unsupervised end-to-end ASR that practically works is not trivial, and has not been fully addressed in previous literature.
>
> > The methodology is not very new. Although there are several differences, some prior studies try to transcribe unseen languages with seen language ASR systems (e.g., Hasegawa-Johnson, Mark A., et al. "ASR for under-resourced languages from probabilistic transcription." IEEE/ACM Transactions on Audio, Speech, and Language Processing 25.1 (2016): 50-63.).
>
> Thanks for the reference. As for most previous work in unsupervised ASR, pointed paper uses ASR with pronunciation dictionaries. In contrast, as stated above, we introduce a practical approach for training end-to-end unsupervised ASR models. In addition, pointed paper i) uses now outdated GMM ASR ii) states that “self-training (ST) [...] costs very little, and benefits little”, while self-training (a.k.a. pseudo-labeling) is a core component of our approach (critical for low WER performance). We will add the reference.
>
> > Did you use SSLs? Since SSLs are obtained by unsupervised training, combining this method and SSL would be very powerful.
>
> Indeed, it has been shown that SSL and pseudo-labeling are complimentary for monolingual semi-supervised ASR settings, see *Xu, Q., et.al. Self-training and pre-training are complementary for speech recognition. ICASSP 2021* and *Zhang, Y., et.al Bigssl: Exploring the frontier of large-scale semi-supervised learning for automatic speech recognition. Journal of Selected Topics in Signal Processing 2022*. We plan to study this in a follow up paper.
>
> > Can you explain why this method uses CTC? Is there a particular reason, or is this method applicable to the other methods (e.g., HMM, attention-based encoder-decoder, RNN-T)? I think that this part makes the paper's scope narrow.
>
> We picked CTC for simplicity. As mentioned above, our method is generalizable to any acoustic (seq2seq, RNN-T...) or language models which can output posterior probabilities (see Eq. 1).

---

> > ### Author Response · Authors · 2023-11-14
> > **Authours Rebuttal [Continue]**
> >
> > > Section 4.1 "(iii) converting characters into the Latin token set via unidecode3 package; characters failing the conversion were discarded from the text": Can you describe the examples? Also, I'm concerned that if we discard some characters in the reference, we cannot evaluate the performance validly with the other reports. Can you clarify this part?
> >
> > Common Voice datasets are released without any text normalization and may contain non-native characters in the transcription. In practice, we found this occurs rarely (< 0.01%). Using unidecode is a way to normalize the datasets in a generalizable manner (across different languages), with little effect on the WER (given those events occur rarely). We will release the normalization procedure for reproducibility. Typical examples: křížová → krízová, 奔熊 → Ben Xiong.
> >
> > > Section 5.7: It is difficult to conclude since they are very different to compare (e.g., I'm not sure which one is more difficult using 800h of labeled German data and 60k hours of unlabeled English data). Can you explain the benefits of your method more clearly?
> >
> > wav2vec-U 2.0 and our approach are difficult to compare as they are very different in nature. We trade (phoneme lexicon, GAN training, and large amount of unlabeled target audio) for (simple pseudo-labeling, limited amount of labeled source audio). In practice, as labeled source audio is always available, our approach is most likely applicable. In contrast, wav2vec-U 2.0 requires a phoneme lexicon and large amount of target audio, which may not always be available.
> >
> > > For me, the method is a little bit over-claimed since this method is primarily applicable to languages with the same scripts or with transliteration systems. It would not be easy to apply this method to the ideogram languages (e.g., Chinese and Japanese, although they are rich-resource languages, and we can build ASR systems easily). I think the paper requires some discussion about it (e.g., adding it to the DISCUSSION AND LIMITATIONS section?).
> >
> > Thanks for your suggestion, we will add this as a discussion. As the point of the paper is to show how and when end-to-end unsupervised ASR (via cross-lingual PL) can work, we did not investigate cases where complex transliteration systems are required: in that case one could instead also rely on phonemes. Using an adequate transliteration system, one could convert Chinese writing hanzi → pinyin (e.g. ”你好“ → ”ni hao“) or Japanese writing kanji → romaji (e.g. ”日本“ → ”Nihon“), and still use our approach.
> >
> > > Section 2.2: suddenly appears. This should be shown around Eq. (1), where first appears.
> >
> > In Eq. (1), $\alpha = 0$ is for greedy decoding and $\alpha > 0$ is for LM beam-search decoding. We will clarify this in the text.
> >
> > > Figure 3 is too small... Please improve it.
> >
> > Detailed plots are available in Appendix (Figures 5, 6, 7, 8). Figure 3 only provides an overview of WER performance across different cross-lingual settings. We will make it bigger.

---

### Official Review · Reviewer_UAMi · 2023-11-10

**Soundness:** 3 good
**Presentation:** 3 good
**Contribution:** 2 fair
**Rating:** 6
**Confidence:** 4

**Summary:**

In this work, the authors propose a cross-lingual unsupervised ASR training framework built on top of the iterative pseudo labeling (IPL) method. They assume a practical situation where unpaired audio and text is available for some low-resource languages and the proposed method is designed to leverage existing source AM (obtained from supervised training on a source language) to generate pseudo labels for the target audio under the regulation of the target LM. The resulting target AM is then iteratively trained on the pseudo labels. Experiments are designed to explore different combination of target & source languages, impact of data size etc. The results demonstrate the effectiveness of the proposed method.

**Strengths:**

1. The design of the experiment is comprehensive. The source & target language set covers not just common European languages but also the less common Arican languages such that the cross-family source & target language situation can also be studied.
2. Comparison with baseline systems such as supervised training and wav2vec-U 2.0 shows that the proposed method is effective in training an ASR model in unsupervised way.

**Weaknesses:**

1. The contribution of this work is limited in cross-lingual scenarios.
2. Baseline system setting is relatively limited. In the core validation experiments (section 5.2), the baseline is only zero-shot evaluation w/ and w/o target LM plus the fully supervised training. It would be more informative if other unsupervised training methods can be compared side-by-side.
3. The paper does not demonstrate sufficient novelty. The author sets an assumption: the target language has no labelled speech accessible but has a fair amount of text data to train an LM. The paper reports how an existing IPL method works under this assumption.

**Questions:**

Question:
1. The LMs seem to be trained and fixed for experiments. Have you tested the correlation between the LM's PPL VS. final WERs?

Suggestion:
1. Please check and fix typos (e.g. section 5.2: "any Indo-European language, any Indo-European language")
2. Please give pointers to the tables/figures in line. E.g. in section 5.5 "the results below" doesn't correctly point to the corresponding table.

---

> ### Author Response · Authors · 2023-11-14
> **Authours Rebuttal**
>
> We would like to thank Reviewer UAMi for their time. Please find below our comments:
>
> > The contribution of this work is limited in cross-lingual scenarios.
>
> > Baseline system setting is relatively limited. In the core validation experiments (section 5.2), the baseline is only zero-shot evaluation w/ and w/o target LM plus the fully supervised training. It would be more informative if other unsupervised training methods can be compared side-by-side.
>
> One of the main contribution of our paper is to show that end-to-end unsupervised ASR is practical (in the sense that it may almost match fully supervised ASR). In contrast, successful previous unsupervised ASR approaches were phoneme-based. We compare with the best available end-to-end unsupervised ASR system, which is wav2vec-U 2.0 on LJSpeech (see Sec. 5.7).
>
> > The paper does not demonstrate sufficient novelty. The author sets an assumption: the target language has no labelled speech accessible but has a fair amount of text data to train an LM. The paper reports how an existing IPL method works under this assumption.
>
> Unsupervised **end-to-end** ASR that practically works is a significant novelty brought by this paper. End-to-end ASR alleviates the need for any phoneme lexicon required in other successful unsupervised ASR approaches (like wav2vec-U 2.0). Another notable difference with wav2vec-U 2.0 is that we do not rely on adversarial training at all. We will highlight better these two points in the paper.
>
> Note also that wav2vec-U 2.0 concludes that “Future work includes simplifying the self-training pipeline and removing the need for a phonemizer.” We demonstrate in this paper a way to do this.
>
> > The LMs seem to be trained and fixed for experiments. Have you tested the correlation between the LM's PPL VS. final WERs?
>
> Correct. We did observe a correlation between LM PPL and WER, in the context of n-gram-based word LMs. We also observed that increasing the dictionary LM size (number of words) improved WER. We will report these results.
>
> > Please check and fix typos (e.g. section 5.2: "any Indo-European language, any Indo-European language")
>
> > Please give pointers to the tables/figures in line. E.g. in section 5.5 "the results below" doesn't correctly point to the corresponding table.
>
> Thanks for spotting this, we will fix it!

---

> > ### Author Response · Authors · 2023-11-23
> > **LM ablation**
> >
> > Dear Reviewer UAMi,
> >
> > Please find below ablation for the $en \rightarrow sw$ cross-lingual PL (greedy decoding) with 2, 3, 4-gram LMs trained with top-200k vocabulary all.
> >
> > | ngram | LM PPL (w/ OOV) test | LM PPL test (w/o OOV) | WER dev | WER test |
> > | ---| ---| ---| ---| ---|
> > | 2 | 436 | 595 | 36.8 | 39.7 |
> > | 3 | 245 | 339 | 25.9 | 28.9 |
> > | 4 | 194 | 270 | 24.1 | 27.1 |

---

### Comment · Area_Chair_Rgne · 2023-11-10
**reviewer-author discussions**

Dear All,

The reviewer-author discussion period will be from Nov. 10 to Nov. 22. For reviewers, please read the authors' responses and acknowledge it, respond to them early on in the discussion, and discuss points of disagreement. Thank you!

AC

---

> ### Comment · Area_Chair_Rgne · 2023-11-21
> **Please discuss authors' response**
>
> Dear Reviewers,
>
> The authors have submitted the response to all reviews. Would you please read the authors' responses and acknowledge it, respond to them early on in the discussion, and discuss points of disagreement? Thank you!
>
> AC

---

### Meta-Review · Area_Chair_Rgne · 2023-12-05

**Metareview:**

The paper proposes a novel method for unsupervised ASR using character-level AMs from other languages to generate pseudo labels for the target language. This is a new and interesting idea that has not been explored before in the area of unsupervised ASR. The paper provides a clear and detailed description of the proposed method and its implementation. The paper also presents experimental results on Common Voice and LJSpeech datasets, showing the effectiveness and superiority of the proposed method over the baseline wav2vec-U 2.0.

The biggest strength is that the paper demonstrated the cross-lingual pesudo labeling strategy works in the multilingual scenario. From the application perspective, the proposed method has impacts.

My major concern of this work is that the novelty is very limited. The paper simply extends the monolingual pseudo labeling (PL) method to the proposed cross lingual pseudo labeling method. Although the authors argued “Unsupervised end-to-end ASR that practically works is a significant novelty brought by this paper.”, the work is more like applying PL into a new problem (unsupervised ASR).

As the experiments mainly focus on the languages with shared Latin alphabets, it is really challenging to predict the performance when the characters of source language and target language are significantly different. Although the authors provided the thought of how to deal with Chinese and Japanese in the rebuttal, there is no result to support whether the thought is right or not. Furthermore, the authors even suggested one possible solution is to leverage phonemes in that challenging situation while in the paper the authors believe characters are better phonemes for the modeling.

The authors stated that they “trade (phoneme lexicon, GAN training, and large amount of unlabeled target audio in wav2vec-U 2.0) for (simple pseudo-labeling, limited amount of labeled source audio).” However, simplicity sometimes restricted the scope of the proposed method. Overall, the paper’s scope is very limited.

It is not surprising that cross lingual modeling helps, and there are large amounts of ASR literature showing it. As pointed out by reviewer UAMi, the zero-shot evaluation has weak baselines without comparing with other methods leveraging cross lingual information. Furthermore, as acknowledged by the authors, self-supervised learning (SSL) is also effective for leveraging unlabeled data, but the authors didn’t build a stronger baseline with SSL.

**Justification For Why Not Higher Score:**

There are too many concerns on this paper such as the novelty and experiment verification. The biggest concern is that the paper has limited novelty which simply extends the  monolingual pseudo labeling (PL) method to the crosslingual scenario.

 Compared to the ICLR 2024 submissions in the speech area, this paper ranks at the bottom.

**Justification For Why Not Lower Score:**

N/A

---

### Decision · Program_Chairs · 2024-01-16

Reject